# R-LAtte: Visual Control via Deep Reinforcement Learning with Attention Network

## Abstract

Attention mechanisms are generic inductive biases that have played a critical role in improving the state-of-the-art in supervised learning, unsupervised pre-training and generative modeling for multiple domains including vision, language and speech. However, they remain relatively under-explored for neural network architectures typically used in reinforcement learning (RL) from high dimensional inputs such as pixels. In this paper, we propose and study the effectiveness of augmenting a simple attention module in the convolutional encoder of an RL agent. Through experiments on the widely benchmarked DeepMind Control Suite environments, we demonstrate that our proposed module can (i) extract interpretable task-relevant information such as agent locations and movements without the need for data augmentations or contrastive losses; (ii) significantly improve the sample-efficiency and final performance of the agents. We hope our simple and effective approach will serve as a strong baseline for future research incorporating attention mechanisms in reinforcement learning and control.

## 1 Introduction

Attention plays a crucial rule in human cognitive processing: a commonly-accepted hypothesis is that humans rely on a combination of top-down selection (e.g., zooming into certain parts in our perception field) and bottom-up disruption (e.g., getting distracted by a novel stimuli) to prioritize information that is most likely to be important for survival or task completion (Corbetta & Shulman L., 2002; Kastner & Ungerleider G., 2000). In machine learning, attention mechanisms for neural networks have been studied for various purposes in multiple domains, such as computer vision (Xu et al., 2015; Zhang et al., 2019a), natural language processing (Vaswani et al., 2017; Brown et al., 2020), and speech recognition (Chorowski et al., 2015; Bahdanau et al., 2016). For example, Zagoruyko & Komodakis (2017) proposed transferring knowledge between two neural networks by aligning the activation (attention) maps of one network with the other. Springenberg et al. (2014) and Selvaraju et al. (2017) proposed a gradient-based method to extract attention maps from neural networks for interpretability. Attention also stands behind the success of Transformers (Vaswani et al., 2017), which uses a self-attention mechanism to model dependencies in long-sequence language data.

However, attention mechanisms have received relatively little attention in deep reinforcement learning (RL), even though this generic inductive bias shows the potential to improve the sample-efficiency of agents, especially from pixel-based environments. More frequently explored directions in RL from pixels are unsupervised/self-supervised learning (Oord et al., 2018; Srinivas et al., 2020; Lee et al., 2020; Stooke et al., 2020; Kipf et al., 2020): Jaderberg et al. (2017) introduced unsupervised auxiliary tasks, such as the Pixel Control task. Srinivas et al. (2020) applied contrastive learning for data-efficient RL and Stooke et al. (2020) further improved the gains using temporal contrast. Another promising direction has focused on latent variable modeling (Watter et al., 2015; Zhang et al., 2019b; Hafner et al., 2020; Sekar et al., 2020; Watters et al., 2019): Hafner et al. (2019) proposed to leverage world-modeling in a latent-space for planning, and Hafner et al. (2020) utilized the latent dynamics model to generate synthetic roll-outs. However, using such representation learning methods may incurs expensive back-and-forth costs (e.g., hyperparameter tuning). This motivates our search in a more effective neural network architecture in the convolutional encoder of an RL agent.

In this paper, we propose R-LAtte: **R**einforcement **L**earning with **Atte**ntion module, a simple yet effective architecture for encoding image pixels in vision-based RL. In particular, the major components of R-LAtte are

- **Two-stream encoding**: We use two streams of encoders to extract non-attentional and attentional features from the images. We define our attention masks by applying spatial Softmax to unnormalized saliencies computed by element-wise product between non-attentional and attentional features. We show that the attentional features contain more task-relevant information (e.g., agent movements) while the non-attentional features contain more task-invariant information (e.g., agent shape).
- **Adaptive scaling**: Once the attention masks are obtained, they are combined with original non-attentional features, and then provided to RL agents. Here, to balance the trade-off between original and attentional features, we introduce a learn-able scaling parameter that is being optimized jointly with other parameters in the encoder.

We test our architecture on the widely used benchmarks, DeepMind Control Suite environments (Tassa et al., 2018), to demonstrate that our proposed module can (i) extract interpretable task-relevant information such as agent locations and movements; (ii) significantly improve the sample-efficiency and final performance of the agents without the need for data augmentations or contrastive losses. We also provide results from a detailed ablation study, which shows the contribution of each component to overall performance.

To summarize, the main contributions of this paper are as follows:

- We present R-LAtte, a simple yet effective architecture that can be used in conjunction for encoding image pixels in vision-based RL.
- We show that significantly improve the sample-efficiency and final performance of the agents on continuous control tasks from DeepMind Control Suite (Tassa et al., 2018).

## 2 RELATED WORK

**Reinforcement learning from pixels**    RL on image-inputs has shown to be benefited from representation learning methods using contrastive losses (Oord et al., 2018; Srinivas et al., 2020; Lee et al., 2020; Stooke et al., 2020; Kipf et al., 2020), self-supervised auxiliary task learning (Jaderberg et al., 2017; Goel et al., 2018; Sekar et al., 2020), and latent variable modeling (Watter et al., 2015; Zhang et al., 2019b; Hafner et al., 2020; Sekar et al., 2020; Watters et al., 2019). Along with those successes, Laskin et al. (2020) and Kostrikov et al. (2020) recently showed that proper data augmentations alone could achieve competitive performance against previous representation learning methods. Different from existing representation learning and data augmentation methods, we focus on architecture improvements specific to RL from pixels, which has been largely uncharted by previous works.

**Attention in machine learning**    Human understand scenes by attending on a local region of the view and aggregating information over time to form an internal scene representation (Ungerleider & G, 2000; Rensink, 2000). Inspired by this mechanism, researchers developed attention modules in neural networks, which directly contributed to lots of recent advances in deep learning especially on natural language processing and computer vision applications (Vaswani et al., 2017; Bahdanau et al., 2015; Mnih et al., 2014; Xu et al., 2015). Using attention mechanism in RL from pixels has also been explored by previous works for various purposes. Zambaldi et al. (2018) proposed self-attention module for tasks that require strong relational reasoning. Different from their work, we study a more computationally-efficient attention module on robot control tasks that may or may not require strong relational reasoning. Choi et al. (2019) is the closest study in terms of the architectural choice. One main difference is in the objective of the study. Choi et al. (2019) investigate the effect of attention on Atari games and focus on the exploration aspect, while we focus on improving the learning of robot controls using the DeepMind Control Suite environments (Tassa et al., 2018). Levine et al. (2016) also use an attention-like module for robot control tasks. There are several architectural design differences (e.g., the addition of a residual connection and a Hadamard product), which we will show are crucial for achieving good performance on control tasks (See Section 5.4).

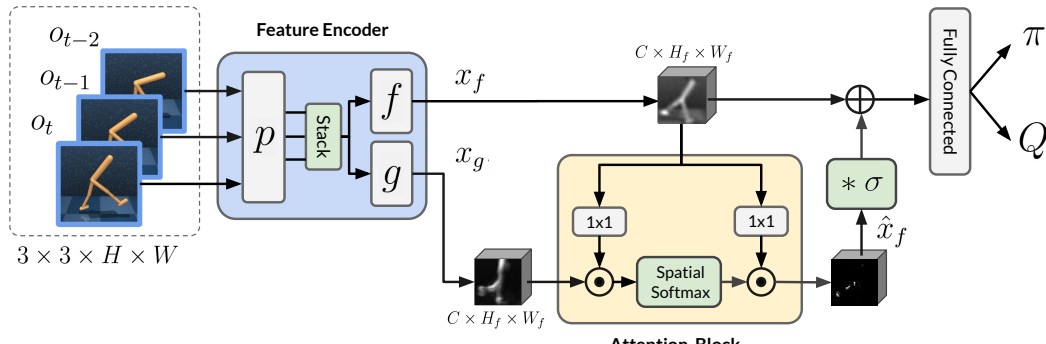

Figure 1: Illustration of our attention-augmented encoder. First, a stack of three image frames are passed as input to a sequence of convolutional layers inside feature encoder (blue box). The feature encoder produces two sets of feature maps: (i) non-attentional features $x_f$ with shape $C \times H_f \times W_f$; (ii) attentional features $x_g$ with shape $C \times H_f \times W_f$. Attention block (orange box) takes in the two sets of feature maps as input, and outputs an attended copy of $x_f$. Non-attentional and attentional features are re-weighted and added together via a residual connection.

## 3 BACKGROUND

**Reinforcement learning** We formulate visual control task as a partially observable Markov decision process (POMDP; Sutton & Barto 2018; Kaelbling et al. 1998). Formally, at each time step $t$, the agent receives a high-dimensional observation $o_t$, which is an indirect representation of the state $s_t$, and chooses an action $a_t$ based on its policy $\pi$. The environment returns a reward $r_t$ and the agent transitions to the next observation $o_{t+1}$. The return $R_t = \sum_{k=0}^{\infty} \gamma^k r_{t+k}$ is the total accumulated rewards from time step $t$ with a discount factor $\gamma \in [0, 1)$. The goal of RL is to learn a policy $\pi$ that maximizes the expected return over trajectories. Following the common practice in DQN (Mnih et al., 2015), the state information from partially observable environments is approximated using stacked input observations, i.e. $s_t$ in equation 1 is approximated by $k$ consecutive frames $o$ (typically $k = 3$): $s_t = (o_t, o_{t-1}, ..., o_{t-k+1})$

**Soft actor-critic** SAC (Haarnoja et al., 2018) is an off-policy actor-critic method based on the maximum entropy RL framework (Ziebart, 2010), which enhances robustness to noise and encourages exploration by maximizing a weighted objective of the reward and the policy entropy. To update the parameters, SAC alternates between a soft policy evaluation and a soft policy improvement. At the policy evaluation step, a soft Q-function, which is modeled as a neural network with parameters $\theta$, is updated by minimizing the following soft Bellman residual:

$$\mathcal{L}_Q(\theta) = \mathbb{E}_{\tau_t \sim \mathcal{B}} \Big[ \big( Q_\theta(s_t, a_t) - r_t - \gamma \mathbb{E}_{a_{t+1} \sim \pi_\phi} \big[ Q_{\bar{\theta}}(s_{t+1}, a_{t+1}) - \alpha \log \pi_\phi(a_{t+1}|s_{t+1}) \big] \big)^2 \Big],$$
(1)

where $\tau_t = (s_t, a_t, r_t, s_{t+1})$ is a transition, $\mathcal{B}$ is a replay buffer, $\bar{\theta}$ are the delayed parameters, and $\alpha$ is a temperature parameter. At the soft policy improvement step, the policy $\pi$ with its parameter $\phi$ is updated by minimizing the following objective:

$$\mathcal{L}_\pi(\phi) = \mathbb{E}_{s_t \sim \mathcal{B}, a_t \sim \pi_\phi} \big[ \alpha \log \pi_\phi(a_t|s_t) - Q_\theta(s_t, a_t) \big].$$
(2)

Here, the policy is modeled as a Gaussian with mean and diagonal covariance given by neural networks to handle continuous action spaces. In this paper, unless mentioned otherwise, all policies are modeled in this way.

## 4 R-LATTE

In this section, we present R-LAtte: **R**einforcement **L**earning with **Atte**ntion module, a simple, yet effective architecture for encoding image pixels in vision-based RL, which typically faces the challenge of partial observability and high-dimensional inputs. By using an attention mechanism, our encoder enforces the agent to select and focus on a subset of its perception field. There are two key components in our architecture (see Figure 1): 1) two-stream encoding, and 2) adaptive scaling.

### 4.1 TWO-STREAM ENCODING FOR ATTENTIONAL FEATURES AND NON-ATTENTIONAL FEATURES

Our two-stream encoding starts by processing each frame in the observation $s_t = [o_t \quad o_{t-1} \cdots \quad o_{t-k+1}]$ independently using a shared encoder $p$. This results in a stacked, encoded observation $s'_t = [p(o_t) \quad p(o_{t-1}) \cdots \quad p(o_{t-k+1})]$. We choose to deviate from the common practice of stacking the frames together before encoding them using a convolutional encoder together (e.g., (Mnih et al., 2013; 2015; Kostrikov et al., 2020; Laskin et al., 2020)) since using a shared encoder for each frame yields better performance in our experiments (See Figure 5 for supporting experimental results).

With the stacked encoded observations, the two streams start to branch out with two convolutional encoders, $f$ and $g$, which encode the stacked observations into non-attentional feature maps $x_f = f(s'_t)$ and attentional feature maps $x_g = g(s'_t)$. Both feature maps $x_f, x_g \in \mathbb{R}^{C \times H \times W}$ have $C$ channels and $H \times W$ spatial dimensions.

We then use a dot product between these two feature maps to produce the soft attention mask as follows:

$$A = \text{Softmax}\left(q(x_f) \odot x_g\right), \tag{3}$$

where $q$ is a $1 \times 1$ convolution and $\odot$ denotes the element-wise product. Here, the element-wise product $q(x_f) \odot x_g$ accentuates where strong signals come from both feature maps, and suppresses those otherwise. This accentuation effect is further normalized using a spatial Softmax function to obtain the soft attention mask.

### 4.2 ADAPTIVE SCALING

Finally, we use the attention maps $A$ to obtain attentional feature maps $\widehat{x}_f \in \mathbb{R}^{C \times H_f \times W_f}$, and provide the agent with a mix of features by adding $\widehat{x}_f$ back to its original with a residual connection:

$$x = x_f + \sigma * \widehat{x}_f \quad \text{where} \quad \widehat{x}_f = h(A \odot v(x_f)), \tag{4}$$

where * denotes multiplying one scalar value to each channel of features in $\widehat{x}_f$; $h$, $v$ are two $1 \times 1$ convolutions, and $\sigma \in \mathbb{R}^C$ are scalars that define the importance of attentional feature maps. $\sigma$ are modeled as learnable network parameters, and get optimized together with other parts of the encoder. In our experiments, they are initialized to be all zeros, but quickly adapted to different values as the training proceeds: intuitively, $\sigma$ controls the amount of non-attentional versus attentional features for the agent to use, and making $\sigma$ learnable allows it to adapt to different features as training proceeds. Ablation set 8(b) provides more experimental results using alternatives to $\sigma$-weights that under-perform using $\sigma$s, and show this need for dynamically control the two-stream features.

## 5 EXPERIMENTS

### 5.1 SETUP

We demonstrate the effectiveness of R-LAtte on a set of challenging visual control tasks from Deep-Mind Control Suite (Tassa et al., 2018), which are commonly used as benchmarks in previous work (Srinivas et al. 2020, Laskin et al. 2020, Kostrikov et al. 2020). This benchmark present a variety of complex tasks including bipedal balance, locomotion, contact forces, and goal-reaching with both sparse and dense reward signals. For evaluation, we consider two state-of-the-art RL methods: Dreamer (Kostrikov et al., 2020), a state-of-the-art model-based RL method that utilizes the latent dynamics model to generate synthetic roll-outs; and DrQ (Kostrikov et al., 2020), a state-of-the-art model-free RL method that applies data augmentations to SAC (Haarnoja et al., 2018). For our method, we train an agent using SAC with R-LAtte encoder, which applies the proposed attention module to the image encoder architecture from SAC-AE (Yarats et al., 2019). We report the learning curves across three runs on various task domains including Hopper, Finger, Walker, HalfCheetah, Ball-in-Cup and record the learning curve. The details of architecture and experimental setups are provided in Appendix B.

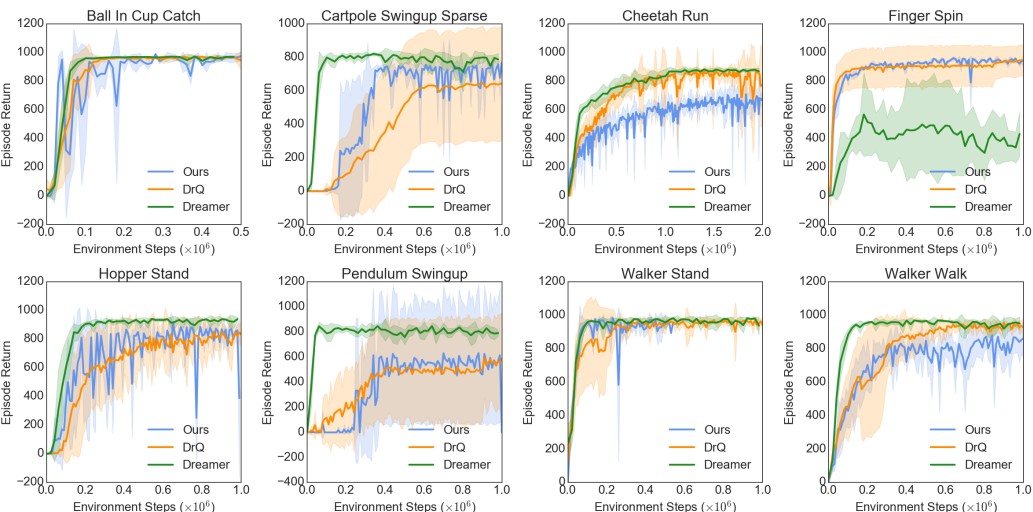

Figure 2: Performances of DeepMind control suite tasks trained with SAC using different encoder architectures for pixel inputs. Our attention network (blue curve) significantly outperforms the baseline encoder, and achieves similar final performance with SAC trained from state inputs.

Figure 3: Learning curves on DeepMind control suite. The solid line and shaded regions represent the mean and standard deviation, respectively, across three runs. Our attention network alone, without any representation learning, world model fitting, or data augmentation techniques, performs on par with the state-of-the-art methods.

## 5.2 MAIN RESULTS

**Comparison with other encoder architectures for pixel input**. We first compare our attention-augmented encoder with a 4-layer (each contains 32 filters) convolution encoder from SAC-AE (Yarats et al., 2019), which is standardized in recent works (Srinivas et al. 2020, Laskin et al. 2020, Kostrikov et al. 2020). For both ours and baseline (denoted by SAC:Pixel), we train agents using SAC without representation learning and data augmentation in order to verify the gains from architecture. Figure 1 shows that our method significantly outperforms SAC:Pixel in terms of both sample-efficiency and final performance. In particular, our method matches the performance of agents trained with state inputs (denoted by State:SAC), which clearly shows the benefits of attention module in RL from pixels.

**Comparative evaluation**. In Figure 3, we compare the performance of our method with the state-of-the-art DrQ and Dreamer. One can note that our attention network can perform on par with the state-of-the-art methods without any representation learning, world model fitting, or data augmentation techniques. We remark that our method has lower variance compared to other baselines in almost all environments.

## 5.3 ANALYSIS

**Supervised setting.** To further verify the effectiveness of the attention module, we test our architecture on an offline supervised learning task. Specifically, we collect replay buffer from a well-train agent that achieves around 1000 average reward and optimize the policy with the attention module

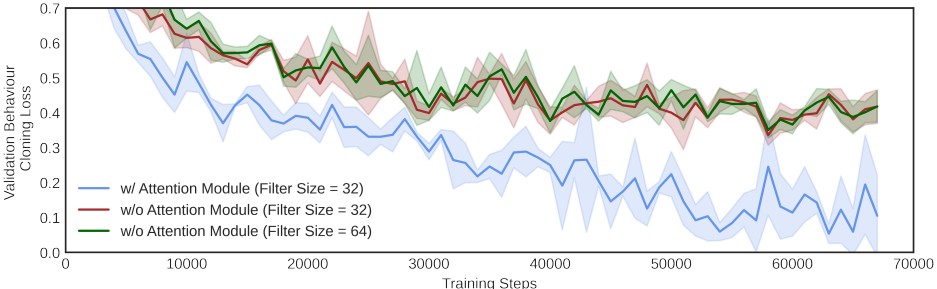

Figure 4: Behaviour cloning performance comparison of our architecture (w/ Attention Module) and the architecture used in RAD (Laskin et al., 2020) (w/o Attention Module). Additionally, we use random cropping data augmentation when training the network with the RAD architecture. Since our architecture have additional encoders that add more parameters, we also include a wider version of the architecture used in RAD (Filter Size = 64) that has more parameters than our architecture (Filter Size = 32) for comparison. Even with random cropping, the architecture used in RAD underperforms our architecture. See more details of the offline experimental setup in Appendix A.

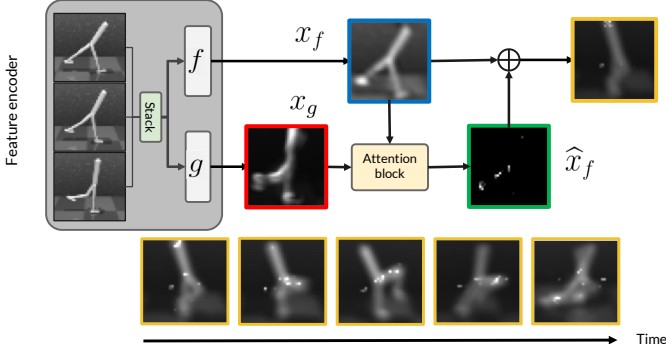

Figure 5: Visualization of outputs from the proposed attention module. Our agent demonstrates a selective feature-extraction scheme that pays more attention to the most relevant locations for each chosen action.

using the behaviour cloning objective (see equation 5 in Appendix A). We compare the validation performance of our architecture against the architecture used in RAD (Laskin et al., 2020) (see Figure 4). Our architecture outperforms the baseline consistently on walker walk task with a faster convergence rate and a better final performance.

**Visualize attention**. To understand the effects of our attention module, we visualize its outputs in Figure 5. First, we remark that two streams of feature outputs, $x_f$ (blue box) and $x_g$ (red box) extract very different information from sample input sources, and the attended output $\widehat{x}_f$ (green box) highlights the activated locations in both features. Also, as shown in the final combined features $x_f + \sigma * \widehat{x}_f$ (yellow box), we find that agent is able to consistently locate itself, captures body movements, and ignores task-irrelevant information in the perception field such as background or floor patterns. Over a course of actions taken, the agent also dynamically changes its attention focus (highlighted by the attention maps) in an interpretable manner, e.g., paying extra attention to its knee joint when taking a stride.

## 5.4 ABLATION STUDIES

The performance gain presented in the above section comes solely from our attention-augmented neural network architecture for encoding pixel images. Here, to provide further insights and justifications for our design, we provide a set of ablation studies on the proposed encoder's major

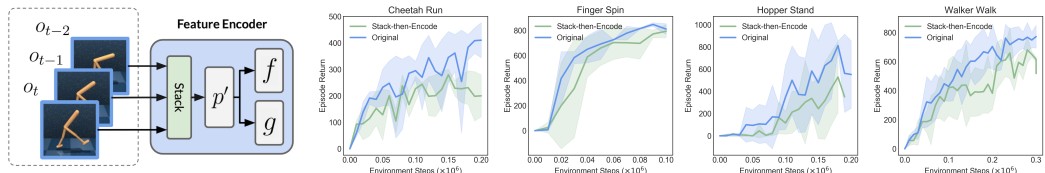

(a) Illustration of the stack-then-encode architecture

(b) Performance comparison between the ablation and our original encoder

Figure 6: We evaluate the ablation encoder $p'$ on a subset of visual control tasks, and demonstrate that, although the original shallow encoder $p$ uses a smaller capacity, it outperforms the ablation encoder consistently across the tested domains.

components. For each ablation, we report the average and standard deviation of learning curves from three ramdom seeds.

## 6 ABLATION ON THE ARCHITECTURE FOR SHARED ENCODER MODULE $g$

**Shared-encoder** $p$. As described in section 4.1, inside our convolutional feature encoder module, we use both a shared encoder $p$ and two parallel encoders $f$ and $g$. In our implementation, $p$ is a shallow, one-layer convolution that, in contrast to common practices of stacking and then encoding consecutive image frames, first encodes each time-step's image observation separately, then stack the features together afterwards. In our implementation, we use a one-layer convolution that, takes in 3 channels (for RGB input) and outputs 10 channels, such that stacking three encoded frames will give 30 total feature channels. For ablations, we swap this convolution layer with a bigger filter size: for a stack of 3 frames, it takes in 9 input channels and output 30. With other parts of the architecture kept the same, we test this stack-then-encode layer on Walker-walk, as shown in Figure 6. With increased filter size, this ablation encoder effectively has a bigger capacity, but as the performance gap suggests, this does not work as well as using a smaller layer, which preserves the temporal changes between consecutive frames.

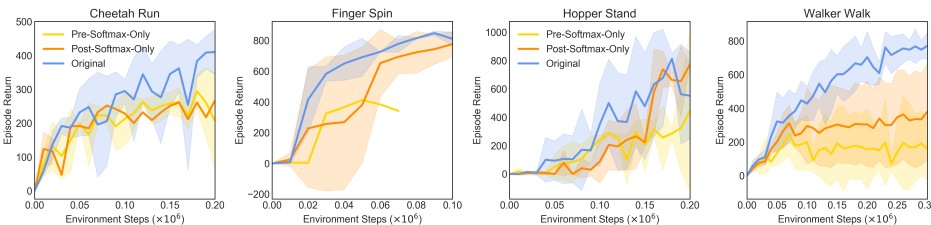

(a) Performance comparison between the ablation attention block and our main architecture

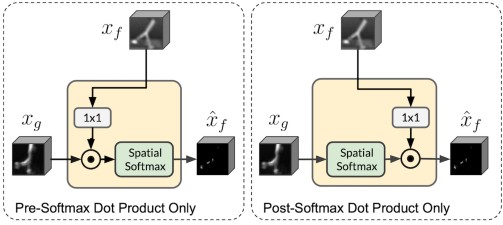

(b) Illustration for the ablation attention blocks that use only one Hadamard product

Figure 7: Ablation illustrations and learning curves on Walker-walk, Cheetah-run, Finger-spin and Hopper-stand.

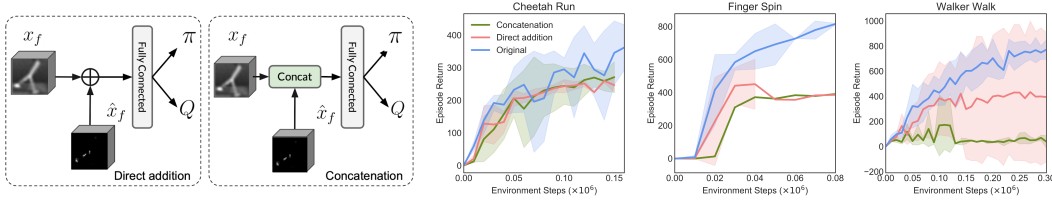

(a) Illustration of two alternative ways of combining $x_f$ and $\widehat{x}_f$

(b) Learning curves

Figure 8: (a) Illustration of direction addition and concatenation, as alternatives to the adaptive scaling method used in our main architecture. (b) Performance comparison between using the ablation combination methods and adaptive scaling using $\sigma$.

**Hadamard products inside the attention block**. Given two inputs $x_f$ and $x_g$ as described in section 4.1, the attention module performs Hadamard product twice, and applies a spatial Softmax function in between. The first Hadamard product can be seen as an element-wise masking of one feature array with another, and the second product is an "attending" step that utilizes the attention maps produced by the spatial Softmax, which we can verify through visualized outputs from the encoder. As ablations, we test two variants of this operation sequence, both does Hadamard product only once, but differs in location, i.e. before or after the Softmax function, as illustrated in Figure 7.

**Adaptive scaling**. Here, with other parts of the encoder architecture fixed, we perform two ablations on alternative ways to combine $x_f$ and $\widehat{x}_f$, namely the non-attentional and attentional features. (A) We experiment a direct residual combination $x_f + \widehat{x}_f$, i.e. without $\sigma$ re-weighting. (B) Instead of addition, we concatenate the two features, and correspondingly increase the input dimension for the subsequent fully connected (FC) layers (output feature dimension for FC layer are kept the same). As the learning curve shows, the two ablations clearly underperforms the original version, which suggests the need for more a flexible, learning-based combination of the features.

## 7 CONCLUSION

We present Reinforcement Learning with Attention module (R-LAtte), a simple yet effective architecture for RL from pixels. On DeepMind Control benchmarks, our simple architecture is able to achieve similar performance compared against representation learning and data augmentation methods, which are currently the best-performing ones in the literature. We also illustrate that our proposed module extracts interpretable task-relevant information without the need for data augmentations or contrastive losses. We hope that our architecture will serve as a simple and strong baseline for future research in RL from pixels, especially for investigations on neural network architectures for encoding high-dimensional inputs. Potential future directions from this work include combining representation learning or data augmentation methods, or both with the attention module, as well as exploring better attention architecture that is more suitable for visual-based RL.

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

# Appendix

## A OFFLINE BEHAVIOUR CLONING EXPERIMENTAL SETUP

We first train a RAD (Laskin et al., 2020) agent to convergence where it achieved around 1000 average reward (in DeepMind Control Suite, the reward is normalized to be between 0 to 1000). Then, we collect 100,000 transitions and store in a replay buffer by rolling out the agent in the environment. After the replay buffer is collected, we train both the architecture used in RAD (as our baseline) and our architecture using stochastic gradient descent with the following behaviour cloning loss:

$$\mathcal{L}_{\mathrm{BC}}(\phi) = -\mathbb{E}_{\tau_t \in \mathcal{B}} \left[ \log \pi_\phi(a_t | s_t) \right] \tag{5}$$

We use a batch size of 128 and Adam optimizer (Kingma & Ba, 2014) with $(\beta_1, \beta_2) = (0.9, 0.999)$, $\epsilon = 10^{-8}$. To determine the best learning rate for each method, we performed a learning rate search over $0.001, 0.0003$ and $0.003$. We found that learning rate of $0.0003$ worked the best for both the baseline and our architecture. We repeat the same experiment for three different seeds and report the mean and the standard deviation.

## B EXPERIMENTAL SETUPS

We use the network architecture in https://github.com/MishaLaskin/rad for our implementation. We show a full list of hyperparameters in Table 1. We will release our code in the camera-ready version.

Table 1: Hyperparameters used for DMControl experiments. Most hyperparameter values are unchanged across environments with the exception of initial replay buffer size, action repeat, and learning rate.

| Hyperparameter | Value |
|---|---|
| Augmentation | Crop |
| Observation rendering | $(100, 100)$ |
| Observation down/upsampling | $(84, 84)$ |
| | 2000 reacher, easy; walker, walk |
| Number of updates per training step | 1 |
| Initial steps | 1000 |
| Stacked frames | 3 |
| Action repeat | 2 finger, spin; walker, walk |
| | 4 reacher, easy |
| | 8 cartpole, swingup |
| Hidden units (MLP) | 1024 |
| Evaluation episodes | 10 |
| Evaluation frequency | 2500 cartpole, swingup |
| | 1000 finger, spin; reacher, easy; walker, walk |
| Optimizer | Adam |
| $(\beta_1, \beta_2) \rightarrow (f_\psi, \pi_\phi, Q_\theta)$ | $(.9, .999)$ |
| $(\beta_1, \beta_2) \rightarrow (\alpha)$ | $(.5, .999)$ |
| Learning rate $(f_\psi, \pi_\phi Q_\theta)$ | $1e - 3$ |
| Learning rate $(\alpha)$ | $1e - 4$ |
| Batch Size | 128 |
| $Q$ function EMA $\tau$ | 0.01 |
| Critic target update freq | 2 |
| Convolutional layers | 4 |
| Number of filters | 32 |
| Non-linearity | ReLU |
| Encoder EMA $\tau$ | 0.05 |
| Latent dimension | 50 |
| Discount $\gamma$ | .99 |
| Initial temperature | 0.1 |

