# OpenReview forum: "R-LAtte: Attention Module for Visual Control via Reinforcement Learning"
_ICLR.cc/2021/Conference — Reject_

### Official Review · AnonReviewer3 · 2020-10-18
**Official Blind Review #3**

**Rating:** 4
**Confidence:** 4

**Review:**

##########################################################################

**Summary**:

This paper proposes to apply the soft attention mechanism in a CNN network to boost the learning speed of an RL agent in environments such as DeepMind Control Suite. The key idea is that the attention mechanism is a better network architecture and can extract interpretable task-relevant information to improve the learning. However, attention has been used in several past RL works. And paper misses some important comparisons. Overall, the contributions of the paper are limited.

##########################################################################

**Strengths**:

The proposed method shows comparable performance against the baselines such as Dreamer and DrQ, and outperforms pure vision-based SAC.

##########################################################################

**Weaknesses**:

The motivation in the introduction section does not seem to be strong. In particular,
the second paragraph seems to be disconnected from the paper. Why do authors spend many words explaining self-supervised/unsupervised learning, model-based learning/planning, etc.?

Using attention in RL is not novel and has been explored in many works [1-5]. For example, [2] also proposed a CNN architecture with a self-attention mechanism. How does the proposed architecture in the paper compare to the one in [2]?

It's also not clear how good the proposed attention block is compared to other attention architectures that are proposed in previous works (self-attention, attention on flattened feature vectors [1], etc.).

The proposed method uses a stack of 3 observations, which can provide the policy more temporal information, while the baselines do not seem to use a stack of historical observations. Hence, it's not clear whether the claim that the proposed method achieves similar performance with SOTA still hold if SOTA algorithms are also given the same amount of information as input.

The keypoint-based method [6] has shown that the network can capture task-relevant information such as keypoint locations. This paper also shows some visualization about the attended regions on the observations in Figure 5. From what both papers show in the visualization, it seems the keypoint-based method can capture better interpretable task-relevant information. It would provide more a more comprehensive view of the series of works if the authors can comment on [6].


The paper compares to SAC with images as input. Such learning is typically slow. [7] shows that using an asymmetric actor and critic can significantly speed up the learning while requiring no state information at test time. How does the attention-based policy compare to this line of works?

**Missing Ablations**:

(1) For the ablation on shared-encoder, the paper provides a comparison between two settings:
* 3 RGB images as input (3 channels as input), each of them are encoded using the same encoder (10 channels for each image output), and the outputs are then stacked (30 channels as output in the end);
* 3 RGB images are stacked first (9 channels as input) and processed by an encoder (30 channels as output). These two settings have a different number of network parameters. It could be the latter one has more parameters and hence takes a bit longer time to train.

To provide a more thorough analysis, one should also compare to the following setting:
* Convert each RGB image into grayscale, and stack the three grayscale images (3 channels as input), and then process them by an encoder that produces an output with 30 channels. While converting RGB images to grayscale lose some color information, such information should not affect the learning or final performance in the environments that the paper experimented with.

Also, I wonder how the curves look after 0.2M or 0.3M steps as shown in Figure (6, 7, 8). Could you provide the training curves for at least 1M training steps for the ablation study figure (Figure 6, 7, 8)?

(2) How does the number of times steps (in the paper, it is 3) in the observation input affect the learning? An ablation on different stacking length will better demonstrate the importance and effect of having a history of observations.

##########################################################################

**Minor points**:

In Section 4.1 (Page 4), it should be Figure 6 rather than Figure 5 to support the claim that using a shared encoder gives better performance.

The structure of the writing is a bit off. Shouldn't section 6 be a subsection for section 5.4?


##########################################################################

**Reference**:

[1] Mishra, Nikhil, et al. "A simple neural attentive meta-learner." arXiv preprint arXiv:1707.03141 (2017).

[2] Manchin, Anthony, Ehsan Abbasnejad, and Anton van den Hengel. "Reinforcement learning with attention that works: A self-supervised approach." International Conference on Neural Information Processing. Springer, Cham, 2019.

[3] Fang, Kuan, et al. "Scene memory transformer for embodied agents in long-horizon tasks." Proceedings of the IEEE Conference on Computer Vision and Pattern Recognition. 2019.

[4] Sorokin, Ivan, et al. "Deep attention recurrent Q-network." arXiv preprint arXiv:1512.01693 (2015).

[5] Mott, Alexander, et al. "Towards interpretable reinforcement learning using attention augmented agents." Advances in Neural Information Processing Systems. 2019.

[6] Kulkarni, Tejas D., et al. "Unsupervised learning of object keypoints for perception and control." Advances in neural information processing systems. 2019.

[7] Pinto, Lerrel, et al. "Asymmetric actor critic for image-based robot learning." arXiv preprint arXiv:1710.06542 (2017).

---

> ### Author Response · Authors · 2020-11-25
> **Response**
>
> >In Section 4.1 (Page 4), it should be Figure 6 rather than Figure 5 to support the claim that using a shared encoder gives better performance. The structure of the writing is a bit off. Shouldn't section 6 be a subsection for section 5.4?
>
> Thank you for pointing out the room for improvement in our writing structure. The previously-mentioned section on unsupervised/model-based methods were intended as background information on recent progress for pixel-based RL that yield SOTA results on the benchmark tasks we evaluated on, and is indeed not a direct motivation for our method since we took a parallel approach by investigating encoder architecture. The section titles are fixed as suggested for our updated version -- apologies for any confusion during paper reading.
>
> >It's also not clear how good the proposed attention block is compared to other attention architectures that are proposed in previous works (self-attention, attention on flattened feature vectors [1], etc.).
>
> There are indeed many works in reinforcement learning that use various kinds of attention in their approach, and we limit ourselves to a pure architecture level and for pixel-based control tasks, hence we compared results directly with SOTA methods on the tested domain. We agree that more detailed comparisons with other attention-based methods would be optimal, and we really appreciate the linked related papers -- we will prioritize comparing with other related architecture choices for future work.
>
> >The proposed method uses a stack of 3 observations, which can provide the policy more temporal information, while the baselines do not seem to use a stack of historical observations.
>
> We also used a stack of 3 frames for the baseline results, apologies for any confusion.
>
> >The keypoint-based method [6] has shown that the network can capture task-relevant information such as keypoint locations. This paper also shows some visualization about the attended regions on the observations in Figure 5. From what both papers show in the visualization, it seems the keypoint-based method can capture better interpretable task-relevant information. It would provide more a more comprehensive view of the series of works if the authors can comment on [6]
>
> The Transporter proposed in [6] is indeed a relevant work. We would like to point out that their major contribution goes to how to conduct unsupervised discovery of keypoints, and the discovered keypoints can later be used for downstream tasks in control. As presented in their 4.Experiment section, the discovered keypoints show gain in Atari game playing and benefits for exploration -- these don't exactly align with the DeepMind control suite task we tested on. We'd like to adapt to other environments like Atari for future work; but for now, one thing to note is how our attention is very task specific, hence may be seen as "supervised" by task rewards, as compared to their "discovered" keypoints that requires some pre-training stage, while ours is trained end-to-end purely based on reward signal.
>
> >The paper compares to SAC with images as input. Such learning is typically slow. [7] shows that using an asymmetric actor and critic can significantly speed up the learning while requiring no state information at test time. How does the attention-based policy compare to this line of works?
>
> Asymmetric actor-critic is indeed a promising direction for image-based control problems, but it has one downside that some form of state information should be obtainable -- hence we are more motivated to study purely pixel-based RL to generalize to more scenarios. We are not aware of any work that investigated asymmetric methods on these DMC suite tasks; given the dense state-variables are indeed available on this simulated environment, we'd expect a better performance than pixel-only approaches.
>
>
> The suggested ablations are also very on-point and we will make sure to consider for future work. Thank you very much for your time and feedback!

---

### Official Review · AnonReviewer2 · 2020-10-26
**Trivial architecture modification with an attention module to inconclusively boost performance in image-based continuous Deep RL**

**Rating:** 4
**Confidence:** 5

**Review:**

The paper proposes an alternative encoder architecture for an image-based deep RL agent that leverages attention. The authors suggest to compute attentional and non-attentional convolutional features, which later are combined together via a weighted (learned) residual connection. The designed architecture seems to improve performance of the agent on standard continuous control tasks from the DeepMind suite.

Significance:
The overall novelty and significance of the paper is low. While I appreciate the authors’ drive to research alternative network architectures in deep RL, I’m unfortunately unable to find any significant insight either from a theoretical (not studied) or empirical (the experimental setup is questionable) perspectives.  More details:

Pros:
 + Interesting investigation into an architecture choice for a deep RL agent. I think this is an important direction to tackle and unfortunately there has been very little attention given to this problem.
 + Interesting ablation that demonstrates the learned attention map and what different features are responsible for.

Cons:
- The novelty, insight, and contribution are limited. Moreover, the empirical evidence provides inconclusive support for the proposed method.
- Weak baselines. In Fig 2 performance of SAC:Pixels is weaker than in the literature (See Fig 6a in https://arxiv.org/pdf/1910.01741.pdf). In Fig 3 performance of DrQ is significantly weaker than reported in the original paper (https://arxiv.org/pdf/2004.13649.pdf) and given that DrQ’s code is publicly available and more importantly fully reproducible (personal experience), this casts doubt on the rigor and validity of conducted experiments.
- This work goes against a common practice in RL and reports performance of their algorithm using only 3 random seeds.
- There is very little information provided about the exact architecture of the attention block, I would appreciate more details here.
- Absence of the source code makes me skeptical that the reported results are reproducible, given that the authors build on a publicly available code base (SAC-AE: https://github.com/denisyarats/pytorch_sac_ae).
- The supervised setting experiment (Sect 5.3) is also inconclusive since the authors compare train loss performance for each of the agents. Obviously agents that use data augmentation would overfit less and thus report higher loss on training dataset. Besides, it is not clear if the dataset for this experiment came from the agent with attention module or not. If so, this will also make learning harder for RAD as it has weaker affinity with the generated data.
- The authors suggest that they don’t use data augmentation, but in Appendix B Table 1 they also suggest that random crops are being used. This needs to be further clarified.

Quality:
The paper’s quality is mediocre. A fully empirical paper requires greater experimental rigor and clarity.

Clarity:
The paper would benefit from providing more transparency over the experimental setup and clear reporting of the used hyper parameters, especially given the simplicity of the proposed method. Several places in the paper (that I detailed above) were either conflicting or ambiguous.

---

> ### Author Response · Authors · 2020-11-25
> **Response**
>
> Thank you for your time and comments!
>
> > Weak baselines. In Fig 2 performance of SAC:Pixels is weaker than in the literature (See Fig 6a in https://arxiv.org/pdf/1910.01741.pdf). In Fig 3 performance of DrQ is significantly weaker than reported in the original paper (https://arxiv.org/pdf/2004.13649.pdf)
>
> We indeed overlooked the results in Figure 6(a) from Yarats et al. and ran the baseline (SAC:Pixel) by ourselves with a smaller batch size. As for DrQ, our plots are from after we contacted the authors and got a copy of their publicly available data from https://github.com/denisyarats/drq/tree/master/data but perhaps the versions are different. We agree that more careful presentation and explanation of our experimental results is needed, but hopefully our message of performing better than baseline and close to SOTA is still consistent. Thank you for pointing us to the related work.
>
> >The supervised setting experiment (Sect 5.3) is also inconclusive since the authors compare train loss performance for each of the agents. Obviously agents that use data augmentation would overfit less and thus report higher loss on training dataset. Besides, it is not clear if the dataset for this experiment came from the agent with attention module or not. If so, this will also make learning harder for RAD as it has weaker affinity with the generated data
>
> First of all, we reported the validation loss rather than the training loss in the supervised experiments. Also, the dataset came from a well-trained RAD agent (the details about the data collection and RAD agent training were included in Appendix A). We specifically chose to use RAD agent for data collection for the reason that the reviewer mentioned. The RAD agent would naturally have stronger affinity to the collected data while still performing worse than our method.
>
> >The authors suggest that they don’t use data augmentation, but in Appendix B Table 1 they also suggest that random crops are being used. This needs to be further clarified
>
> Thank you for pointing this out -- it's a typo due to directly copying the hyper-parameters table over. We didn't use data augmentation for the presented results.

---

### Official Review · AnonReviewer1 · 2020-10-27
**Interesting approach to add an attention module to deep RL, but some issues should be addressed.**

**Rating:** 5
**Confidence:** 3

**Review:**


###################################

Summary: This paper aims to combine visual attention mechanisms to deep RL by proposing a simple attention module in the convolutional encoder of the deep RL agent. Two-stream encodings and adaptive scaling of balance between non-attentional/attentional masks are major components of this module. The authors present empirical results of their algorithm and compares with SOTA baselines in deep mind control suite.

###################################

Pros

1. This paper is fairly well written and easy to follow. One of the main contributions of the attention module is two-stream encoding, where there are separate non-attentional features and attentional features extractor. Attentional features have more task-specific information, while non-attentional features have task-agnostic information. This attention module can be combined with any RL baseline algorithm, like SAC. The structure of attention module is simple and easy to understand.

2. R-Latte shows competitive performance in deepmind control suite. R-Latte performs much better than SAC+pixel, and almost achieves as good performance with SAC+state. When compared with SOTA algorithms (DrQ and Dreamer), R-Latte shows comparable performance in many domains.

3. Visualization of attention module is helpful. As shown in Figure 5, the attended output \hat{x_f} (green box) captures relevant activated locations, and the agent can dynamically change the attention focus depending on each action.

###################################

Cons & Questions

Major

1. Adaptive scaling of \sigma: the \sigma is an alternative parameter that should be tuned to balance between attentional and non-attentional features. I think the role of this parameter is consequential, but the paper lacks detailed description about this parameter. Could you plot how \sigma changes over the course of training? (\sigma is C-dimensional vector and initialized as zero, so you could plot the average of \sigma over the course of training, and how it evolves over time.) Considering that adaptive scaling of balance between two encoder is one of the two major contributions of this work, the paper lacks sufficient explanations on this part.

2. Need more justification for shared encoder for consecutive frames:  one of the difference of this encoder from other conventional encoders is the use of shared-encoder p. The original algorithms stack the consecutive frames first, and then put them altogether into the encoder; whereas in this work, each frame is encoded by p, and then stacked after the encoding. Although the difference between stack-first-baselines and your algorithm is shown in the ablation studies (Figure 6), I don’t see reasonable explanations on why this would benefit the performance. Do you have any intuition of why separate-encoding-then-stacking performs better? To my knowledge, there is a reason in stacking consecutive frames first (e.g. in DQN), because stacking consecutive frames contains its own information (Mnih et al. 2015) and separating them might break those information.

Minor

3. The performance of R-Latte is comparable with SOTA (Dreamer, DrQ), but doesn’t outperform them. In most case, R-Latte performs slightly worse than DrQ or Dreamer; it’s totally fine, but I think you should change the phrase like “significantly improve sample-efficiency and final performance of the agents”.

4. Does your method show lower variances than SOTA baselines as mentioned in page 5? Only looking from the graph plots, I don't see that. Could you support this claim with statistical/numerical values?


####################

Typos & citation suggestions

- In section 5.1. setup, I think you cited a wrong paper for Dreamer; you cited Kostrikov et al. 2020 for both Dreamer and DrQ. But you should fix it to Hafner et al. 2018 for Dreamer.
- Consider citing these works at the related work section and adding some discussions about them too:

[1] Mott, A.; Zoran, D.; Chrzanowski, M.; Wierstra, D, Rezende, D. J. 2019. Towards interpretable reinforcement learning using attention augmented agents. In Advances in Neural Information Processing Systems, 12350–12359.

[2] Greydanus, S.; Koul, A.; Dodge, J.; Fern, A. 2018. Visualizing and understanding atari agents. In International Conference on Machine Learning, 1792–1801.

[3] Yuezhang, L.; Zhang, R.; and Ballard, D. H. 2018. An initial attempt of combining visual selective attention with deep reinforcement learning. arXiv preprint arXiv:1811.04407


##########################################

Overall, I think this is a nice work, but given the concerns I have above and marginal/incremental advances in empirical results, I think this paper doesn't yet meet the threshold of ICLR. However, if these concerns are addressed, I am happy to raise my score after the rebuttal period. Also, related work section can be improved by covering more past works: there are a lot of works on visual attention/saliency, and there have been some recent works attempting to use attention for deep RL agents (not just the ones that I mentioned above).

---

> ### Author Response · Authors · 2020-11-25
> **Response**
>
> Thank you for your time and the very detailed questions/comments!
>
> >1. ... Considering that adaptive scaling of balance between two encoder is one of the two major contributions of this work, the paper lacks sufficient explanations on this part.
>
> We are adding a histogram of evolving sigma values to the appendix, and we observe that although initialized to be all zeros, the values at each index change into different non-zero ranges, suggesting different emphasis on each channel of the encoded features. We agree that the paper lacks more detailed explanation on the training of sigma values, and would like to investigate more on this for follow-up work.
>
> >2. Need more justification for shared encoder for consecutive frames: one of the differences of this encoder from other conventional encoders is the use of shared-encoder p.
>
> We designed this encode-then-stack step with the intention to preserve the changes between each frame, so that the the two streams of attentional and non-attentional encoding can select from features at each time-step, and decide their own way to stack those features. We think that because all the later layers in both encoders use stacked encoding, our method is still similar to an all-stack encoding for the purpose of extracting information between time-steps -- as the ablation results suggest, stack-first encoding does not lead to a huge gap in performance.
>
> >3. The performance of R-Latte is comparable with SOTA (Dreamer, DrQ), but doesn’t outperform them. In most case, R-Latte performs slightly worse than DrQ or Dreamer; it’s totally fine, but I think you should change the phrase like “significantly improve sample-efficiency and final performance of the agents”.
>
> Thank you for pointing this out. We intended to claim the significant gain over only the baseline results that don't use other techniques (the gray curves in Figure 2), but the wording is perhaps misleading and we will fix it in the updated version.
>
> >4. Does your method show lower variances than SOTA baselines as mentioned in page 5? Only looking from the graph plots, I don't see that. Could you support this claim with statistical/numerical values?
>
> The 'lower variance' claim indeed calls for more justification and we'd like to take that down until more experimental results come in. The SOTA results we compare to were averaged over more seeds and used larger batch sizes, and we are yet to establish a fair numerical comparison for now.

---

### Author Response · Authors · 2020-11-25
**Thank you for the review and we would like to improve for future work**

Thank you to the reviewers for the insightful feedback and comments. Towards the goal of improving image-based reinforcement learning, our investigation on attention mechanisms applied to deep neural network architectures was acknowledged for the gain over baseline methods. We believe the use of attention mechanisms, and more broadly, the question of designing deep neural network architectures for reinforcement learning, are very interesting and important problems to keep pursuing. Hence we appreciate the reviewers’ constructive questions and reviews for the current progress, and for future work, we would prioritize the suggested directions including investigation on details in this architecture, gathering more experimental results from both ours and other relevant image-based RL methods, etc.

---

### Decision · Program_Chairs · 2021-01-07
**Final Decision**

**Decision:**

Reject

**Comment:**

This paper proposes an attention-endowed architecture for deep image-based RL. While some positive points were raised by the reviewers, most comments were on the negative side.
The reviewers noted marginal/incremental advances in terms of empirical results and low novelty and significance. Moreover, the provided baselines seem weak.
Because of this, the present submission unfortunately does not meet the publication bar.
I recommend the authors take into account the constructive feedback from reviews and discussion and submit an improved version to another venue.